# Slice Sampling Reparameterization Gradients

**David M. Zoltowski**
Princeton Neuroscience Institute
Princeton University
Princeton, NJ 08540
zoltowski@princeton.edu

**Diana Cai**
Department of Computer Science
Princeton University
Princeton, NJ 08540
dcai@cs.princeton.edu

**Ryan P. Adams**
Department of Computer Science
Princeton University
Princeton, NJ 08540
rpa@princeton.edu

## Abstract

Many probabilistic modeling problems in machine learning use gradient-based optimization in which the objective takes the form of an expectation. These problems can be challenging when the parameters to be optimized determine the probability distribution under which the expectation is being taken, as the naïve Monte Carlo procedure is not differentiable. Reparameterization gradients make it possible to efficiently perform optimization of these Monte Carlo objectives by transforming the expectation to be differentiable, but the approach is typically limited to distributions with simple forms and tractable normalization constants. Here we describe how to differentiate samples from slice sampling to compute *slice sampling reparameterization gradients*, enabling a richer class of Monte Carlo objective functions to be optimized. Slice sampling is a Markov chain Monte Carlo algorithm for simulating samples from probability distributions; it only requires a density function that can be evaluated point-wise up to a normalization constant, making it applicable to a variety of inference problems and unnormalized models. Our approach is based on the observation that when the slice endpoints are known, the sampling path is a deterministic and differentiable function of the pseudo-random variables, since the algorithm is rejection-free. We evaluate the method on synthetic examples and apply it to a variety of applications with reparameterization of unnormalized probability distributions.

## 1 Introduction

Probabilistic modeling is a powerful approach to inferring latent structure in complex real-world processes, but often presents computational challenges for inference and further downstream tasks. In many modern probabilistic models, inference is recast as optimization of a probabilistic objective that takes the form of an expectation of a loss function with respect to some distribution. A salient example is variational inference [27, 3], where a simpler distribution is optimized to approximate a posterior distribution by minimizing the Kullback-Leibler (KL) divergence to the truth. In particular, Monte Carlo methods for estimating the gradient of the KL with respect to the parameters have extended the use of variational inference to non-conjugate Bayesian models [43, 44] and neural networks [31]. Such probabilistic objectives also appear in a number of other applications in machine learning and computational science, including the generator loss in generative adversarial networks [15], computing the sensitivity of expectations under the posterior distribution to prior

35th Conference on Neural Information Processing Systems (NeurIPS 2021).

hyperparameters [21, 19, 12], computing the Black-Scholes delta in computational finance [13], and choosing a design that maximizes the probability of improvement in an experiment [56].

Typically, the probabilistic objectives are comprised of expectations that cannot be computed in closed form, so gradients are often estimated via Monte Carlo sampling [39]. Two popular classes of Monte Carlo gradient estimators are the *score function* estimator [32, 14, 55, 43, 44] and the *pathwise* (or "reparameterization gradient") estimator [23, 24, 45, 31, 49]. Score function gradient estimators are general-purpose, as they apply when the underlying density is differentiable and can be sampled from, even if the cost function is not differentiable. However, they often have high variance and are used with variance reduction techniques (e.g., Ranganath et al. [44]). In contrast, reparameterization gradients apply when the loss function is differentiable and samples from the underlying density can be generated by a known deterministic, differentiable transformation of samples from a simpler distribution that does not depend on the model parameters. Typically, reparameterization gradients have lower variance than score function gradients, provided the loss function is sufficiently smooth [39]. Therefore, developing effective reparameterized gradient estimators has been an active area of research in sensitivity and perturbation analysis [23, 24, 13] and stochastic backpropagation [31, 45].

However, reparameterization gradients are primarily limited to distributions with tractable normalizing constants. This precludes their use for complex models of interest such as energy-based models and non-conjugate Bayesian models (although alternative training methods exist, see e.g. Lawson et al. [33]). Thus, generalizing reparameterization with MCMC to unnormalized distributions is an important direction for developing effective gradient estimators for complicated models. Indeed, some recent work has focused on reparameterization gradients for unnormalized distributions in specialized problems, including Gibbs samplers with reparameterizable conditional sampling steps [54] and dynamics-based MCMC without accept/reject steps (e.g., Salimans et al. [50], Dai et al. [8]). However, developing estimators for general unnormalized distributions using reparameterized gradients and MCMC is not straightforward with existing approaches. For instance, not all Gibbs sampling steps are reparameterizable using current methods, due to, e.g., rejection sampling [11] or "Metropolis-within-Gibbs" sampling [7]. Next, dynamics-based MCMC samplers without accept/reject steps are approximate samplers with asymptotic bias. Finally, a key obstacle in applying MCMC methods with accept/reject steps, such as the Metropolis-Hastings (MH) [22], Metropolis-Adjusted Langevin (MALA) [16], and Hamiltonian Monte Carlo (HMC) [42] algorithms, is that they do not have differentiable sample paths.

An appealing alternative to MCMC methods currently used for reparameterized gradients is slice sampling [41], an auxiliary-variable MCMC method that can be applied to unnormalized probability distributions and does not require an accept/reject step or sensitive tuning parameters. Crucially, the lack of an accept/reject step leads to the key observation that for a fixed pseudo-random sequence, the realized slice sampling Markov chain is differentiable with respect to the model parameters. In this work, we develop reparameterization gradients for samples generated from slice sampling that apply to distributions known only up to a normalizing constant. Slice sampling reparameterization gradients are broadly applicable to complicated multivariate distributions, such as energy-based models (EBMs) [35] and non-conjugate Bayesian models. While the generated samples are correlated and the gradient estimates are biased because we simulate from a finite Markov chain, we demonstrate the efficacy of slice sampling reparameterization gradients in simulations, investigating the bias and variance properties of reparameterized slice sampling in comparison with existing gradient estimators. We then show applications of slice sampling reparameterized gradients to a number of problems in machine learning and statistics in the areas of deep generative modeling, approximate inference, and Bayesian sensitivity analysis.

## 2   Background

In this work we are concerned with probabilistic objectives that take the form of an expectation of a real-valued function $\ell$ with respect to a base density $p_\theta(x)$, i.e.,

$$\mathcal{L}(\theta) = \mathbb{E}_{p_\theta(x)}[\ell(x)]. \tag{1}$$

Typically the expectation cannot be computed in closed form, so computing the gradient of this objective with respect to the distributional parameters $\theta$ requires stochastic estimates of the gradient of the expectation. Two popular classes of stochastic gradient estimators are score function gradients and reparameterization (or pathwise) gradients; see Mohamed et al. [39] for a review.

## 2.1 Continuous reparameterization gradients

Reparameterization gradients are used for problems with a differentiable loss function $\ell$ and continuous density $p_\theta$. They apply when samples from $p_\theta$ can be generated by a deterministic transformation $f_\theta$ of samples from a base distribution $p(\epsilon)$ that does not depend on $\theta$. That is, if $\epsilon \sim p(\epsilon)$ then $x = f_\theta(\epsilon) \sim p_\theta(x)$. Applying this transformation, the gradient of the objective with respect to $\theta$ can then be expressed as an expectation with respect to a distribution that does not depend on $\theta$. Assuming that the derivative and integral can be interchanged [39, Section 4.3.1]:

$$\nabla_\theta \mathcal{L}(\theta) = \nabla_\theta \mathbb{E}_{p(\epsilon)}[\ell(f_\theta(\epsilon))] = \mathbb{E}_{p(\epsilon)}[\nabla_\theta \ell(f_\theta(\epsilon))]. \tag{2}$$

Monte Carlo estimates of the gradient are then computed using samples from $p(\epsilon)$.

Many examples of reparameterization gradients for continuous distributions exist. "One-liner" reparameterization gradients use a simple function to transform the base distribution into the desired parameterized distribution [31, 45, 39]. The inverse CDF (quantile function) can be used to transform uniform random variables into arbitrary 1D distributions, and gradients can be computed when evaluating and differentiating the inverse CDF are numerically tractable. Additional examples are implicit reparameterization gradients [10], doubly reparameterized gradients [53], reparameterization of accept/reject sampling [40], and generalized reparameterization gradients [48].

## 2.2 Markov chain Monte Carlo via slice sampling

For complicated target distributions, direct sampling via, e.g., the inverse CDF, is often not feasible and moreover it is often the case that the target density function can only be evaluated up to an unknown normalization constant; this commonly occurs in Bayesian posterior inference where the marginal likelihood is unavailable. Markov chain Monte Carlo (MCMC) algorithms address both of these challenges. An MCMC algorithm is a recipe for constructing a Markov chain that is easy to simulate and that converges in distribution to the target, gaining computational tractability at the cost of initialization bias and correlation between the samples.

The most common recipe for constructing an MCMC transition operator is the Metropolis–Hastings algorithm [38], which combines an easy-to-simulate proposal distribution with a randomized accept/reject decision that depends only on a ratio of densities, thereby avoiding the need to know the normalization constant. Variations on this theme include Gibbs sampling, in which the proposal is chosen to be the conditional distribution for a subset of the variables, and Hamiltonian Monte Carlo (HMC) [42], in which the proposal is constructed from a fictive dynamical system.

Slice sampling [41] is an alternative approach to MCMC based on the observation that samples taken uniformly from the volume beneath the target PDF have the correct marginal distribution. Slice sampling defines a Markov transition operator that leaves this uniform distribution invariant, and is example of an *auxiliary variable* MCMC method (along with HMC, Swendsen-Wang [51], pseudo-marginal MCMC [1], and others). It is appealing because such transition operators can be constructed without having to identify a proposal distribution and the updates never get "stuck" as can happen with Metropolis–Hastings; it is also generally robust to choices of tuning parameters.

Consider a distribution with density $p_\theta(\boldsymbol{x}) = \frac{1}{Z(\theta)} \pi_\theta(\boldsymbol{x})$ where the normalizing constant may not be known. Slice sampling typically proceeds in two steps. Starting from a point $\boldsymbol{x}_n$, a height $y_{n+1}$ is sampled uniformly beneath the density at $\boldsymbol{x}_n$ such that $y_{n+1} \sim \mathcal{U}(0, \pi_\theta(\boldsymbol{x}_n))$. The height $y_{n+1}$ defines a "slice" through the probability density given by $S = \{\boldsymbol{x} : y_{n+1} < \pi_\theta(\boldsymbol{x})\}$. The next point $\boldsymbol{x}_{n+1}$ is then sampled uniformly from the set $S$. There are different methods for sampling uniformly from $S$, including component-wise updates and construction of hyper-rectangles; here we use *random-direction slice sampling* to sample from the set $S$ (37, Chapter 29.7). A random direction $\boldsymbol{d}$ is generated from a uniform distribution over directions, and the direction defines a line segment through the slice with endpoints $\boldsymbol{x}^-$ and $\boldsymbol{x}^+$. Finally, a value $\boldsymbol{x}_{n+1}$ is sampled uniformly between the endpoints. The most common procedure for finding the endpoints, as proposed by Neal [41], is to use a reversible "stepping-out" procedure followed by "interval shrinking" to determine $\boldsymbol{x}_{n+1}$. In this work we will not use this stepping out procedure and will instead perform a direct search for the slice boundaries. This direct search is key to the present work, as it eliminates the non-differentiable accept/reject step that otherwise appears in most implementations of slice sampling.

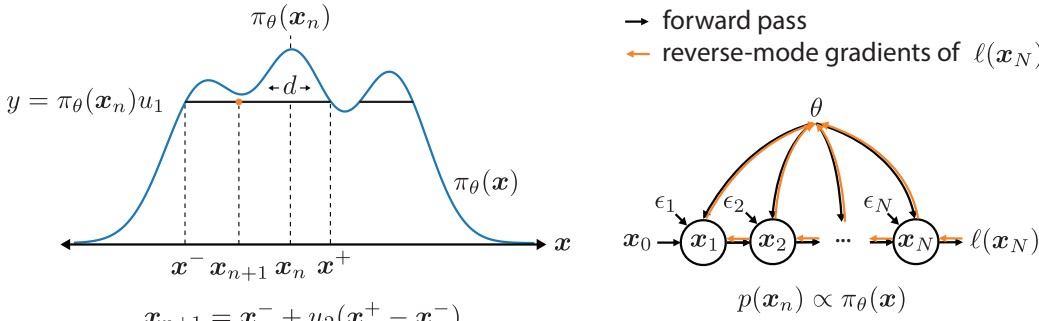

Figure 1: Slice sampling. *Left*: One step of a 1D slice sampler. *Right*: Reparameterized slice sampling computational graph with reverse mode gradients shown for a loss computed on the final sample.

## 3   Slice sampling reparameterization gradients

### 3.1   Random-direction slice sampling with numerical slice endpoints

Here we describe the steps of random-direction slice sampling in more detail. We consider a continuous, unnormalized density $\pi_\theta(\boldsymbol{x})$ with parameters $\theta$ and start from a point $\boldsymbol{x}_n \in \mathbb{R}^d$. We sample three random quantities: two uniform random numbers $u_1, u_2 \in [0, 1]$ and a uniform random unit vector $\boldsymbol{d} \in \mathbb{R}^d$. The value $u_1$ determines the height of the slice $u_1\pi_\theta(\boldsymbol{x}_n)$. The direction $\boldsymbol{d}$ induces a line through the slice. This line intersects with the density at points where $\pi_\theta(\boldsymbol{x}_n + \alpha\boldsymbol{d}) = u_1\pi_\theta(\boldsymbol{x}_n)$ for scalar values $\alpha$. The intersecting points on the slice $S$ induce the set of $\alpha$ values

$$\mathcal{A} := \{\alpha \in \mathbb{R} : \pi_\theta(\boldsymbol{x}_n + \alpha\boldsymbol{d}) = u_1\pi_\theta(\boldsymbol{x}_n)\}. \tag{3}$$

Define the slice endpoints representing the closest intersecting locations on the slice $S$ in the positive and negative directions where $\pi_\theta(\boldsymbol{x}_n + \alpha\boldsymbol{d}) = u_1\pi_\theta(\boldsymbol{x}_n)$ as

$$\boldsymbol{x}^+ := \boldsymbol{x}_n + \alpha^+\boldsymbol{d}, \quad \boldsymbol{x}^- := \boldsymbol{x}_n + \alpha^-\boldsymbol{d}, \qquad \text{where} \quad \alpha^+ := \min_{\alpha>0}\mathcal{A}, \quad \alpha^- := \max_{\alpha<0}\mathcal{A}. \tag{4}$$

The points $\alpha^+$ and $\alpha^-$ are identified using numerical root finding (Appendix A). The next sample $\boldsymbol{x}_{n+1}$ is then taken uniformly between the endpoints $\boldsymbol{x}^-$ and $\boldsymbol{x}^+$:

$$\boldsymbol{x}_{n+1} = \boldsymbol{x}^- + u_2(\boldsymbol{x}^+ - \boldsymbol{x}^-) = \boldsymbol{x}_n + u_2\alpha^+\boldsymbol{d} + (1 - u_2)\alpha^-\boldsymbol{d}. \tag{5}$$

Importantly, $\alpha^-$ and $\alpha^+$ are implicit functions of $\boldsymbol{x}_n$, $\boldsymbol{d}$, $u_1$, and $\theta$, denoted by $\alpha^+(\boldsymbol{x}_n, \boldsymbol{d}, u_1, \theta)$, and $\alpha^-(\boldsymbol{x}_n, \boldsymbol{d}, u_1, \theta)$. Via Equation (5), the next sample iterate $\boldsymbol{x}_{n+1}$ is then a deterministic function of the previous iterate $\boldsymbol{x}_n$, the parameters $\theta$, and the random variables $u_1$, $u_2$, and $\boldsymbol{d}$:

$$\boldsymbol{x}_{n+1} = \boldsymbol{x}_n + u_2\,\alpha^+(\boldsymbol{x}_n, \boldsymbol{d}, u_1, \theta)\,\boldsymbol{d} + (1 - u_2)\,\alpha^-(\boldsymbol{x}_n, \boldsymbol{d}, u_1, \theta)\,\boldsymbol{d}. \tag{6}$$

In Figure 1 and later sections, we use $\epsilon$ to refer to the random draws $u_1$, $u_2$, and $\boldsymbol{d}$. Notably, the per-iteration cost of identifying the slice endpoints is similar for one-dimensional and higher-dimensional distributions, as the algorithm searches for slice endpoints along a single direction per-iteration regardless of the dimensionality of the sampling space.

### 3.2   Differentiating the slice sampling path

Recall that the goal is to estimate the gradient of an expectation taken with respect to $p_\theta(\boldsymbol{x}) \propto \pi_\theta(\boldsymbol{x})$ with an unknown normalization constant. Consider the following reparameterization of the probabilistic objective $\mathcal{L}(\theta)$ and its Monte Carlo estimator $L(\theta)$, respectively:

$$\mathcal{L}(\theta) = \mathbb{E}_{p_\theta(\boldsymbol{x})}[\ell(\boldsymbol{x})] = \mathbb{E}_{p(\epsilon)}[\ell(\boldsymbol{x}(\epsilon; \theta))] \tag{7}$$

$$\approx \frac{1}{N}\sum_{n=1}^{N}\ell(\boldsymbol{x}(\epsilon_n; \theta)) =: L(\theta), \quad \text{where } \epsilon^{(n)} \overset{\text{iid}}{\sim} p(\epsilon),$$

and where $\boldsymbol{x}(\epsilon; \theta)$ is a deterministic function of $\epsilon$ and $\theta$ and $p(\epsilon)$ is the distribution of the uniform random variables $u_1$ and $u_2$ and the uniform random direction $\boldsymbol{d}$. Given an initial sample $\boldsymbol{x}_0$, the slice sampling algorithm generates the samples $\boldsymbol{x}_{1:N} := (\boldsymbol{x}_n)_{n=1}^N$, where $\boldsymbol{x}_n = \boldsymbol{x}(\epsilon^{(n)}; \theta)$ is the value of the function established in Equation (5) evaluated at the previous point $\boldsymbol{x}_{n-1}$ and the sample $\epsilon^{(n)} = (u_1^{(n)}, u_2^{(n)}, \boldsymbol{d}^{(n)})$. After the forward pass of generating the samples, we evaluate the objective function and use reverse-mode automatic differentiation (AD) to compute gradients [17, Section 3.2]. In what follows, we derive the reverse-mode gradients, and for notational simplicity, we will drop the explicit dependence of $\boldsymbol{x}_n$ on $\epsilon$ and $\theta$. Our derivation uses implicit differentiation to efficiently compute gradients of the slice endpoints, avoiding the need to compute gradients through the numerical root finding algorithm.

Applying the chain rule, the gradient of $L(\theta)$ with respect to $\theta$ is given by a recurrence relation backwards through the samples $\boldsymbol{x}_n$:

$$\nabla_\theta L(\theta) = \sum_{n=1}^N [\mathcal{J}_\theta(\boldsymbol{x}_n)]^\mathsf{T} \nabla_{\boldsymbol{x}_n} L, \tag{8}$$

where $\mathcal{J}_\theta(\boldsymbol{x}_n) \in \mathbb{R}^{d \times m}$ is the Jacobian of $\boldsymbol{x}_n$ with respect to $\theta \in \mathbb{R}^m$. Here we observe that the gradient $\nabla_\theta L(\theta)$ depends on the gradients of the loss with respect to each sample, $\nabla_{\boldsymbol{x}_n} L \in \mathbb{R}^d$. For the final sample $\boldsymbol{x}_N$ this value is simply $\nabla_{\boldsymbol{x}_N} L = \frac{1}{N} \nabla_{\boldsymbol{x}_N} \ell(\boldsymbol{x}_N)$. Due to the sequential dependency, for earlier samples $\boldsymbol{x}_n$ where $n = 1, ..., N-1$ the loss gradients need to be backpropagated via

$$\nabla_{\boldsymbol{x}_n} L = \frac{1}{N} \nabla_{\boldsymbol{x}_n} \ell(\boldsymbol{x}_n) + [\mathcal{J}_{\boldsymbol{x}_n}(\boldsymbol{x}_{n+1})]^\mathsf{T} \nabla_{\boldsymbol{x}_{n+1}} L. \tag{9}$$

The computational graph demonstrates how each sample $\boldsymbol{x}_{n+1}$ depends on the previous samples $\boldsymbol{x}_{1:n}$ and the parameters $\theta$ (Figure 1). Therefore, to compute gradients with respect to the parameters and/or the initial condition, the Jacobians $\mathcal{J}_{\boldsymbol{x}_n}(\boldsymbol{x}_{n+1})$ and $\mathcal{J}_\theta(\boldsymbol{x}_{n+1})$ are needed. Using Equation (5) for $\boldsymbol{x}_{n+1}$, these Jacobians can be computed indirectly via adjoints, i.e.,

$$\mathcal{J}_{\boldsymbol{x}_n}(\boldsymbol{x}_{n+1}) = \mathcal{J}_{\boldsymbol{x}_n}\big(\boldsymbol{x}_n + \boldsymbol{d}u_2\alpha^+ + \boldsymbol{d}(1-u_2)\alpha^-\big) = \boldsymbol{I}_d + u_2\boldsymbol{d}\nabla_{\boldsymbol{x}_n}[\alpha^+]^\mathsf{T} + (1-u_2)\boldsymbol{d}\nabla_{\boldsymbol{x}_n}[\alpha^-]^\mathsf{T}$$

$$\mathcal{J}_\theta(\boldsymbol{x}_{n+1}) = \mathcal{J}_\theta\big(\boldsymbol{x}_n + \boldsymbol{d}u_2\alpha^+ + \boldsymbol{d}(1-u_2)\alpha^-\big) = u_2\boldsymbol{d}\nabla_\theta[\alpha^+]^\mathsf{T} + (1-u_2)\boldsymbol{d}\nabla_\theta[\alpha^-]^\mathsf{T}. \tag{10}$$

We use implicit differentiation to compute $\nabla_{\boldsymbol{x}_n}[\alpha^+], \nabla_{\boldsymbol{x}_n}[\alpha^-], \nabla_\theta[\alpha^+], \nabla_\theta[\alpha^-]$. The values $\alpha$ are solutions to

$$f(\boldsymbol{x}_n, \boldsymbol{d}, \alpha, \theta) = \ln \pi_\theta(\boldsymbol{x}_n + \alpha\boldsymbol{d}) - \ln u_1 - \ln \pi_\theta(\boldsymbol{x}_n) = 0. \tag{11}$$

Applying implicit differentiation (17, Chap. 15) results in:

$$\nabla_{\boldsymbol{x}_n}\alpha = -\frac{\nabla_{\boldsymbol{x}_n} f}{\partial f / \partial \alpha} = -\frac{\nabla_{\boldsymbol{x}_n} \ln \pi_\theta(\boldsymbol{x}_n + \alpha\boldsymbol{d}) - \nabla_{\boldsymbol{x}_n} \ln \pi_\theta(\boldsymbol{x}_n)}{\boldsymbol{d}^\mathsf{T} \nabla_{\boldsymbol{x}_n} \ln \pi_\theta(\boldsymbol{x}_n + \alpha\boldsymbol{d})}$$

$$\nabla_\theta\alpha = -\frac{\nabla_\theta f}{\partial f / \partial \alpha} = -\frac{\nabla_\theta \ln \pi_\theta(\boldsymbol{x}_n + \alpha\boldsymbol{d}) - \nabla_\theta \ln \pi_\theta(\boldsymbol{x}_n)}{\boldsymbol{d}^\mathsf{T} \nabla_{\boldsymbol{x}_n} \ln \pi_\theta(\boldsymbol{x}_n + \alpha\boldsymbol{d})}. \tag{12}$$

After computing $\{\nabla_{\boldsymbol{x}_1} L, \ldots, \nabla_{\boldsymbol{x}_N} L\}$, the gradient $\nabla_\theta L$ is then formed by computing by Equation (8). Importantly, we compute $\nabla_\theta L$ without ever fully representing either of the Jacobians by using vector-Jacobian products (Appendix D). We implemented[1] the forward sampling and reverse mode AD in JAX [4].

### 3.3 Gradient of expected log likelihood

In applications, we often require the gradient of the expected log likelihood $\mathbb{E}_{q_\phi(\boldsymbol{x})}[\log q_\phi(\boldsymbol{x})]$ with respect to $\phi$. For example, this arises in variational inference when optimizing the ELBO or KL divergence. For a distribution $q_\phi(\boldsymbol{x}) = \frac{1}{Z(\phi)} e^{f_\phi(\boldsymbol{x})}$, the gradient of the reparameterized objective is

$$\nabla_\phi \mathbb{E}_{q_\phi(\boldsymbol{x})}[\log q_\phi(\boldsymbol{x})] = \nabla_\phi \mathbb{E}_{p(\epsilon)}[\log q_\phi(\boldsymbol{x}(\phi, \epsilon))] \tag{13}$$

$$= \mathbb{E}_{p(\epsilon)}[\nabla_\phi f_\phi(\boldsymbol{x}(\phi, \epsilon)) - \nabla_\phi \log Z(\phi)]. \tag{14}$$

---

[1]Our implementation is available at https://github.com/PrincetonLIPS/slicereparam

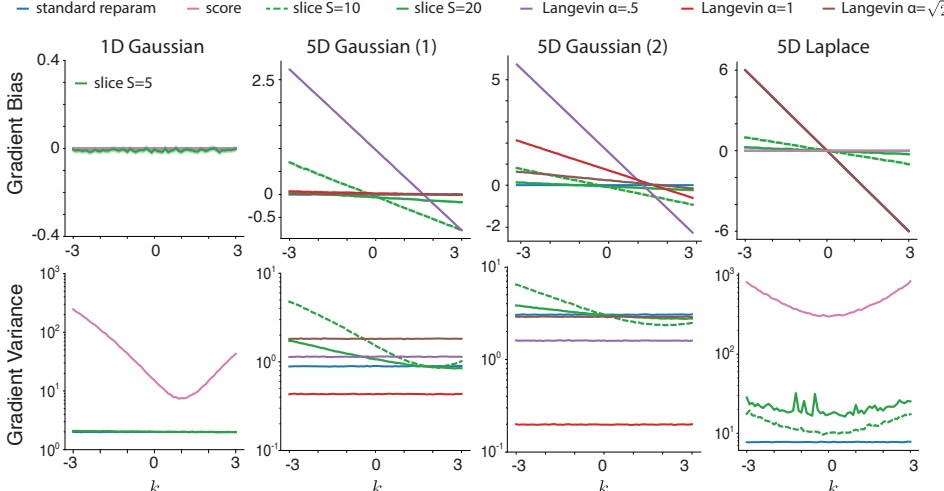

Figure 2: Gradient estimator bias and variance for various Monte Carlo gradient estimators across four different distributions (columns). Slice sampling reparameterization gradients are in green.

This appears to require the gradient of the unknown log normalizing constant. However, applying the total derivative [46] yields

$$\nabla_\phi \mathbb{E}_{p(\epsilon)}[\nabla_\phi \log q_\phi(\boldsymbol{x}(\phi, \epsilon))] = \mathbb{E}_{p(\epsilon)}[\nabla_{\boldsymbol{x}} \log q_\phi(\boldsymbol{x}(\phi, \epsilon))\nabla_\phi \boldsymbol{x}(\phi, \epsilon) + \nabla_\phi \log q_\phi(\boldsymbol{x})] \quad (15)$$

$$= \mathbb{E}_{p(\epsilon)}[\nabla_{\boldsymbol{x}} f_\phi(\boldsymbol{x}(\phi, \epsilon))\nabla_\phi \boldsymbol{x}(\phi, \epsilon)] + \mathbb{E}_{p(\epsilon)}[\nabla_\phi \log q_\phi(\boldsymbol{x})] \quad (16)$$

$$= \mathbb{E}_{p(\epsilon)}[\nabla_{\boldsymbol{x}} f_\phi(\boldsymbol{x}(\phi, \epsilon))\nabla_\phi \boldsymbol{x}(\phi, \epsilon)]. \quad (17)$$

where we dropped the dependence of $\boldsymbol{x}$ on $\phi$ to indicate evaluation at a value $\boldsymbol{x}$. We drop the second term in (16) because it is the expected score function and is therefore equal to zero. Notably, this gradient of the expected log likelihood *does not* require the gradient of the log normalizer and we can compute the necessary quantities to construct Monte Carlo gradient estimates. In our experiments, we use this estimator when optimizing the ELBO and KL divergence with unnormalized models.

## 4 Experiments

Here we apply slice sampling reparameterization gradients to several problems, demonstrating the generality of the approach and the potential appeal of reparameterization gradients for unnormalized distributions. For additional experimental details see the supplementary material (Section C).

### 4.1 Bias and variance

Following Mohamed et al. [39], we quantify the empirical bias and variance of slice sampling reparameterization gradients in comparison to standard reparameterization gradients, score function gradients, and reparameterized Langevin dynamics (Figure 2). We estimate $\nabla_\theta \mathbb{E}_{q(\boldsymbol{x};\theta)}[\frac{1}{D}\sum_i (\boldsymbol{x}_i - k)^2]$ with $\boldsymbol{x} \in \mathbb{R}^D$ for multiple distributions $q$ and various values of $k$. These experiments show that slice sampling gradient estimates have initialization bias for finite sample sizes, which is expected. Importantly, however, we also find that slice sampling reparameterization gradients can be lower variance than score function gradients computed from exact samples from the distribution.

The first example is a one-dimensional Gaussian with $q(x; \theta) = \mathcal{N}(x; \theta, 1)$ and $\theta = 1$. In this case, slice sampling mixes well and the resulting reparameterization gradients are nearly unbiased with variance similar to the standard reparameterization estimator. We next focus on comparing slice sampling with Langevin dynamics on multi-dimensional examples. We consider two 5-dimensional multivariate Gaussian distributions $q(\boldsymbol{x}; \theta) = \mathcal{N}(\boldsymbol{x}; \theta \mathbf{1}, \Psi_i)$ with diagonal covariances $\Psi_1$ and $\Psi_2$ and a Laplace distribution $q(x; \theta) = \prod_i \text{Laplace}(x_i; \theta, b_i)$. While Langevin with a tuned step size may outperform slice sampling in some cases, we find that the performance of Langevin dynamics

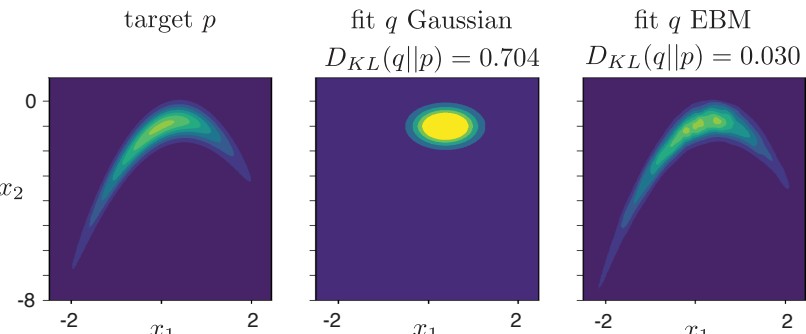

Figure 3: The target $p$ (*left*) and fit mean-field Gaussian (*middle*) and EBM (*right*) approximating distributions. Each panel is shown with the same color scale.

varies considerably for the same set of step sizes across the two multivariate Gaussian distributions. On the other hand, slice sampling performance is consistent across each of the examples and improves with more samples. The versatility of slice sampling is highlighted by the Laplace example. In this case the gradient of the mean from reparameterized Langevin dynamics is zero. On the other hand, slice sampling reparameterization gradients have low bias and notably have substantially lower variance than the score function gradients computed from exact samples from the distribution.

## 4.2 Minimizing KL-divergence

Minimizing the KL-divergence $D_{KL}(q_\theta||p)$ w.r.t. $\theta$ between a parameterized distribution $q_\theta$ and a target $p$ is a core problem of approximate inference. Here we optimize this objective using slice sampling reparameterization gradients on a number of synthetic examples. Our goal for these experiments is simply to demonstrate successful optimization using slice sampling reparameterization gradients, and we expect other methods such as normalizing flows or alternative gradient estimators would also perform well (see Section C.2 for a comparison to a normalizing flow model). We present one experiment in the main text with additional experiments in Section C.7.

To demonstrate the potential utility of unnormalized approximate distributions, we choose a 2D "banana" distribution that has a complicated dependency between the two dimensions as the target (Figure 3). We optimize an EBM approximate distribution

$$q_\phi(\boldsymbol{z}) = \frac{1}{Z(\phi)} \exp(f_\phi(\boldsymbol{z})) \mathcal{N}(\boldsymbol{z}; \mu_\phi, \sigma_\phi^2), \tag{18}$$

where $f_\phi$ is a feed-forward neural network mapping from the two-dimensional input space to a scalar energy. The Gaussian term is included in the model to ensure integrability. We optimized $\theta$ by minimizing $D_{KL}(q_\theta||p)$ using slice sampling reparameterization gradients and Adam [30].

The fit and target distributions are shown in Figure 3 along with a comparison to fitting a mean-field Gaussian to the same objective. Qualitatively, the fit EBM appears remarkably similar to the target. We estimated the KL divergence between the target $p$ and the approximate distributions $q$ using numerical integration. We find that the KL divergence between the fit EBM and the target approaches zero and is much lower than the KL divergence between the target and the mean-field Gaussian posterior (Figure 3).

## 4.3 Variational contrastive divergence with fully reparameterized gradients

Combining variational inference with MCMC methods is an active area of research [50, 26, 3, 47]. One approach is to use samples from a variational posterior to initialize an MCMC chain targeting the true posterior. The distribution of the resulting samples will be closer in KL divergence to the true posterior than the variational posterior, making the ELBO tighter. However, this approach can be challenging because the dependence on the variational parameters decreases as the length of the MCMC chain increases and the distribution of the generated samples is not available in closed form.

Ruiz and Titsias [47] propose the variational contrastive divergence (VCD) objective for combining variational inference with MCMC to address these two problems. Let the approximate posterior be $q_\theta(\boldsymbol{z})$, and let $q_\theta^{(t)}(\boldsymbol{z})$ be the distribution given by sampling an initial value $\boldsymbol{z}^{(0)} \sim q_\theta(\boldsymbol{z})$ and then

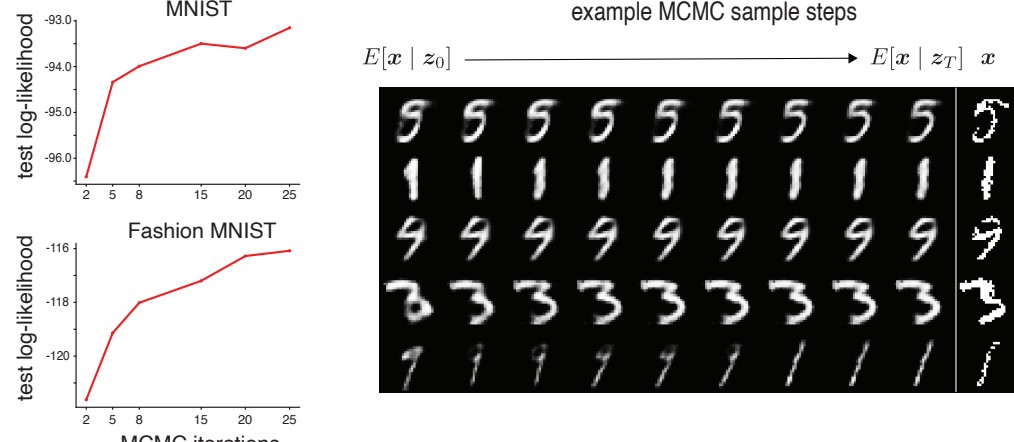

Figure 4: (Left) held out log-likelihood for MNIST and Fashion MNIST as a function of the number of MCMC iterations. (Right) The mean observation conditioned on sample $z_i$ for $z_0$ sampled from the variational posterior and for $T$ samples targeting the true posterior.

running $t$ steps of an MCMC chain targeting the true posterior $p(z \mid x)$. Then the VCD objective is

$$\mathcal{L}_{\text{VCD}}(\theta) = -\mathbb{E}_{q_\theta(z)}[\log p(x, z) - \log q_\theta(z)] + \mathbb{E}_{q_\theta^{(t)}(z)}[\log p(x, z) - \log q_\theta(z)]. \qquad (19)$$

Notably, this does not require evaluating $q_\theta^{(t)}$ and instead only requires sampling from it. However, the second term of the VCD objective is an expectation with respect to MCMC samples, and Ruiz and Titsias [47] use a score function estimator to estimate this gradient. With reparameterized slice sampling we can use fully reparameterized gradients to optimize the VCD objective. Notably, this is an example that requires a differential MCMC algorithm to obtain reparameterization gradients.

We fit a variational autoencoder (VAE) [31, 45] using the VCD objective to MNIST [34] and Fashion MNIST [58]. Our network architectures matched those used in Ruiz and Titsias [47]: the latent dimensionality was 10 and both the encoder and decoder had two hidden layers with

Table 1: Estimated heldout log likelihoods.

| method | MNIST | Fashion MNIST |
|---|---|---|
| VCD [47] | -95.86 | **-117.65** |
| VCD w/ Reparam (Ours) | **-93.99** | -118.02 |

200 units and ReLU nonlinearities. The model was optimized for 400K iterations using ADAM with batch sizes of 100. We also estimated the held-out log-likelihood via the importance sampling approach used in Ruiz and Titsias [47].

In Table 1, we compare the held-out log likelihoods on MNIST and Fashion MNIST for our method with slice sampling reparameterization gradients and the results reported in Ruiz and Titsias [47] for $S = 8$ samples. We find that slice sampling reparameterization gradients perform comparably or better than the original VCD results, with each method having a higher test log likelihood on one of the datasets. As a function of the number of MCMC iterations (Figure 4), the test log likelihood for slice sampling reparameterization gradients appears to scale better than the results obtained in Ruiz and Titsias [47] (Figure 2 in that paper). Finally, for samples generated from the slice sampler, optimization using reparameterization gradients largely outperformed optimization using score function gradients (Appendix C).

We also demonstrate that the learned model and slice sampling procedure provide a meaningful result. In Figure 4, we show the expected value of the observation given a sample from the variational posterior and how 8 steps of slice sampling towards the target posterior improve the latent sample. The resulting conditional mean estimate for $z_T$ appears more similar to the true observation $x$ than the estimate for $z_0$. For example, in the bottom row the initial sample appears more like a nine but is transformed to a one, which is the true observation.

### 4.4 Conditional EBM approximate posterior

We next used slice sampling reparameterization gradients to fit a VAE with a conditional EBM approximate posterior:

$$q(\boldsymbol{z} \mid \boldsymbol{x}) = \frac{1}{Z(\theta, \boldsymbol{x})} \exp\{f_\phi(\boldsymbol{x})^\top g_\phi(\boldsymbol{z}) + \log p_0(\boldsymbol{z})\}. \tag{20}$$

The observations are $\boldsymbol{x}$, the latent variables are $\boldsymbol{z} \in \mathbb{R}^D$, and $\log p_0(\boldsymbol{z})$ is a prior. The functions $f_\phi(\boldsymbol{x})$ and $g_\phi(\boldsymbol{z})$ map to vectors with an embedding dimensionality $D_e$. This differs from the standard VAE in that the encoder $f$ does not map to the mean of the latent space, but rather a separate embedding that is nonlinearly combined by $g(\boldsymbol{z})$ to give the energy. The form of the conditional EBM has been studied previously in Khemakhem et al. [28, 29]. We fit the model to MNIST data (Figure 5) with $D = 20$ and $D_e = 50$. Unconditional samples from the generative model look reasonable. We visualized the embedding of held out test images by the function $f$ using t-SNE [36]. Interestingly, we see clusters in the embedding space corresponding to digits, with some overlap for similar digits. Finally, as a baseline comparison we also fit the model using reparameterized Langevin dynamics. The resulting model provided a qualitatively similar embedding of the digits (Appendix C.4).

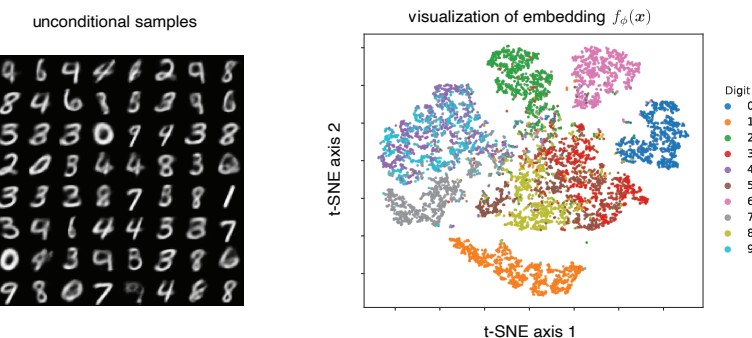

Figure 5: Unconditional samples (*left*) and embedding visualization (*right*) of a VAE fit to MNIST with an EBM approximate posterior.

### 4.5 Adversarial training of EBMs

We next apply our method to training generative adversarial networks (GANs) [15] with EBM generators. Let the generator distribution be $p_\theta(\boldsymbol{x}) \propto e^{f_\theta(\boldsymbol{x})}$. Standard GANs define an implicit distribution over the data points, whereas an EBM defines an unnormalized log-likelihood in the data space. We optimize the Wasserstein GAN [2] objective with the gradient penalty from [18]:

$$\min_\theta \max_{D \in \mathcal{D}} \mathbb{E}_{p_d(\boldsymbol{x})}[D(\boldsymbol{x})] - \mathbb{E}_{p_\theta(\boldsymbol{x})}[D(\boldsymbol{x})] + \lambda \, \mathbb{E}_{p(\hat{\boldsymbol{x}})}[(\|\nabla_{\hat{\boldsymbol{x}}} D(\hat{\boldsymbol{x}})\|_2 - 1)^2] \tag{21}$$

where $\hat{\boldsymbol{x}}$ are sampled uniformly between pairs of real and generated data points [18], $D$ is the discriminator, and $\mathcal{D}$ is the set of 1-Lipschitz functions. We trained two dimensional EBMs using this approach on three synthetic datasets, and we find that the distribution of generated samples from the learned energy qualitatively match the training distribution (Figure 6, *left*). Our network architecture and optimizer were based on those in [52, 6], and in future work we are interested in exploring applications of reparameterization gradients for unnormalized distributions to discriminator-driven latent sampling [6] and latent optimization for GANs [57].

### 4.6 Bayesian sensitivity analysis

The sensitivity of expectations under Bayesian posteriors to prior hyperparameters is a common quantity of interest in Bayesian statistics [21, 19]. One formalization of this sensitivity to the prior is as the gradient of the expectation with respect to the prior hyperparameters. Formally, the *local sensitivity* of an expectation $\mathbb{E}_{p_\alpha}[g(\theta)]$ at a point $\alpha_0$ is $\nabla_\alpha \mathbb{E}_{p_\alpha}[g(\theta)]|_{\alpha=\alpha_0}$, where $p_\alpha$ is the posterior $p_\alpha = p(\theta \mid \mathcal{D}, \alpha)$ [20]. Typically the normalizing constant of the posterior is unknown and score function gradient approaches are needed to estimate this gradient. However, slice sampling reparameterization gradients may provide a lower variance estimator of the sensitivity.

**Baseball example**   We consider a non-conjugate hierarchical model of baseball hit probabilities [9, 5]. The model of the number of successes (hits) for player $j$ in $K_j$ attempts (at bats) is

$$\phi \sim U[0,1] \quad \kappa \sim \text{Pareto}(1, \alpha) \quad \theta_j \sim \text{Beta}(\phi \cdot \kappa, (1-\phi) \cdot \kappa) \quad y_j \sim \text{Binomial}(K_j, \theta_j), \tag{22}$$

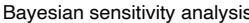

Adversarially-trained EBMs          Bayesian sensitivity analysis

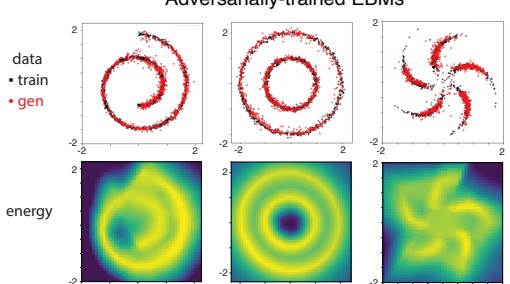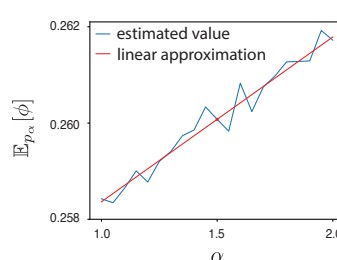

Figure 6: *Left*: Training and learned generative samples (top) and learned energy function (bottom) for the Swiss roll, two circles, and spirals datasets. *Right*: Linear approximation to posterior mean hierarchical hit probability based on the estimated sensitivity at $\alpha = 1.5$.

where $\phi$ determines the mean and $\kappa$ roughly determines the variance of the hierarchical mean hit probability across players. The hyperparameter of interest, $\alpha$, determines the shape of the prior distribution on $\kappa$. We estimated the local sensitivity $\nabla_\alpha \mathbb{E}_{p_\alpha}[\phi]$ using slice sampling reparameterization gradients. We find that the estimated local sensitivity provides a good linear approximation to how the value of $\alpha$ determines the hierarchical mean hit probability (Figure 6, *right*). We present additional results in the supplementary material (Appendix C).

## 5 Discussion

**Limitations**    The slice sampler used in this paper samples random directions from the uniform distribution over directions for identifying one-dimensional slices in higher dimensions. For complex distributions, as the dimensionality increases it may be decreasingly likely that the sampled random direction helps to efficiently explore the distribution. This can reduce the sampling efficiency of the algorithm and limit the performance in higher dimensions. In future work, we are interested in exploring non-uniform distributions over random directions.

Next, the algorithm relies on identifying the closest points that satisfy the slice boundary condition, which are the slice endpoints. To identify these points, we first "step out" from the current point until a bracketing point is found before running the root finder algorithm (Appendix A). Ideally, this bracket contains only one point that satisfies the boundary condition, which is the slice endpoint. However, for multimodal distributions this is not guaranteed, and therefore finding the slice boundaries can be inexact for multimodal distributions. In particular, this is more challenging when modes are closely connected such that there are multiple nearby points that satisfy the slice endpoint condition. We can check if the next sampled point is on the slice to verify if this is an issue, and modify the stepping out procedure to probe more finely spaced locations if needed.

**Conclusion**    In this work we developed slice sampling reparameterized gradients and demonstrated the method on a variety of applications involving approximate inference, energy-based models, GANs, and Bayesian statistics. Among many possible directions, in future work we are interested in combining reparameterized slice sampling with normalizing flows to improve the geometry of the sampling space [25], developing reparameterization gradients for other classes of slice samplers, and continuing to explore applications of reparameterization gradients of unnormalized distributions.

**Societal Impact**    The potential negative societal impacts of our work are broadly the harm that can be done using reparameterization gradient estimators to optimize models. One example would be optimizing GANs for harmful applications such as "DeepFakes."

## Acknowledgments and Disclosure of Funding

We thank Dougal Maclaurin for valuable discussions. Additionally, we thank members of the Princeton Laboratory for Intelligent Probabilistic Systems and the anonymous reviewers for valuable feedback. This work was partially funded by NSF IIS-2007278. D. Zoltowski was supported in part by NIH grant T32MH065214. D. Cai was supported in part by a Google Ph.D. Fellowship in Machine Learning.

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
