# Supplementary Material: Slice Sampling Reparameterization Gradients

**David M. Zoltowski**
Princeton Neuroscience Institute
Princeton University
Princeton, NJ 08540
zoltowski@princeton.edu

**Diana Cai**
Department of Computer Science
Princeton University
Princeton, NJ 08540
dcai@cs.princeton.edu

**Ryan P. Adams**
Department of Computer Science
Princeton University
Princeton, NJ 08540
rpa@princeton.edu

## A    Implementation details

As mentioned in the main text, we implemented the forward slice sampling with numerical root finding and reverse mode automatic differentiation in JAX [1]. Here we describe components of our implementation in more detail.

**Random direction**    The random direction $\boldsymbol{d}$ is sampled by simulating $d$ standard normal random variables and then normalizing.

**Root finding**    The forward sampling process for the reparameterized slice sampling algorithm requires numerical root finding to identify the slice endpoints. This corresponds to finding the $\alpha^+$ and $\alpha^-$ that satisfy

$$\pi(\boldsymbol{x}_n + \alpha\boldsymbol{d}; \theta) - u_1\pi(\boldsymbol{x}_n \, ; \theta) = 0. \tag{1}$$

Importantly, as mentioned above, we are interested in two specific roots: the roots $\alpha^- \, < 0$ and $\alpha^+ > 0$ that are closest to zero.

We used a standard bisection algorithm to identify these scalar roots. This corresponded to running two root finding operations; one bracketed below zero $\alpha^- \, \in \, [-a, \delta]$ and one bracketed above zero $\alpha^+ \in [\delta, a]$ where $\delta$ is a small positive constant. Notably, for multimodal distributions there may be multiple roots inside the brackets, and the bisection algorithm will only be guaranteed to find one of these roots (rather than the root closest to zero). Therefore we first identified the bracket value $a$ with a stepping out procedure. We did this by stepping out $a$ logarithmically until the bracketing condition was satisfied. Once the condition was satisfied, we proceeded with the bisection algorithm.

We wrote the root finder using custom code in JAX such that it could be compiled using `jit` and batched using `vmap`. These two features are critical for speed. The `jit` compilation makes the root finding fast, and `vmap` makes it easy and fast to batch the root finding operation across multiple independent sampling chains.

**Forward sampling**    The forwards sampling process takes as input random noise variables, the current parameters, and a function that computes the log probability of the distribution of interest (up to an additive constant). It proceeds with the slice sampling steps, using the root finding implementation described in the previous subsection to identify the roots. We also implemented this function to be compiled using `jit` and batched across multiple chains using `vmap`.

35th Conference on Neural Information Processing Systems (NeurIPS 2021).

**Reverse mode gradients**  We implemented the backwards pass using the efficient computations described in D. Again, we implemented this function to be compiled using `jit` and batched across multiple chains using `vmap`.

## B  ELBO Gradient Derivation

Here we provide a detailed derivation of the gradient of the ELBO with an unnormalized energy-based model. Let the approximate posterior be given by an unnormalized density

$$q_\phi(\boldsymbol{z}) = \frac{1}{Z(\phi)} e^{f_\phi(\boldsymbol{z})}. \tag{2}$$

The reparameterized ELBO is

$$\mathcal{L}(\theta, \phi) = \mathbb{E}_{q_\phi(\boldsymbol{z})}[\log p_\theta(\boldsymbol{x}, \boldsymbol{z}) - \log q_\phi(\boldsymbol{z})] \tag{3}$$

$$= \mathbb{E}_{p(\epsilon)}[\log p_\theta(\boldsymbol{x}, \boldsymbol{z}(\phi, \epsilon)) - \log q_\phi(\boldsymbol{z}(\phi, \epsilon))] \tag{4}$$

where $\boldsymbol{z}(\phi, \epsilon)$ is a sample generated from the slice sampler with unnormalized density $f_\phi(\boldsymbol{z})$ and the noise $\epsilon$ are the set of random variables $u_1$, $u_2$, and $\boldsymbol{d}$ used to generate the reparameterized samples. The Monte Carlo estimate of the gradient given a sample $\epsilon$ is

$$\hat{\nabla}(\epsilon, \phi) = \nabla_\phi[\log p_\theta(\boldsymbol{x}, \boldsymbol{z}(\phi, \epsilon)) - \log q_\phi(\boldsymbol{z}(\phi, \epsilon))] \tag{5}$$

$$= \nabla_\phi[\log p_\theta(\boldsymbol{x}, \boldsymbol{z}(\phi, \epsilon)) - f_\phi(\boldsymbol{z}(\phi, \epsilon)) + \log Z(\phi)]. \tag{6}$$

This appears to require the gradient of the normalizing constant $Z(\phi)$. We can estimate this gradient using samples generated from $q_\phi(\boldsymbol{z})$ via slice sampling. However, we take a different approach. Applying the total derivative [6], we have

$$\hat{\nabla}_{\text{TD}}(\epsilon, \phi) = \nabla_{\boldsymbol{z}}[\log p_\theta(\boldsymbol{x}, \boldsymbol{z}(\phi, \epsilon)) - f_\phi(\boldsymbol{z}(\phi, \epsilon)) + \log Z(\phi)]\nabla_\phi \boldsymbol{z}(\phi, \epsilon) \tag{7}$$

$$+ \nabla_\phi[\log p_\theta(\boldsymbol{x}, \boldsymbol{z}) - \log q_\phi(\boldsymbol{z})]. \tag{8}$$

The components $\nabla_{\boldsymbol{z}} \log Z(\phi)$ and $\nabla_\phi \log p_\theta(\boldsymbol{x}, \boldsymbol{z})$ are zero. Next, we can ignore $\nabla_\phi \log q_\phi(\boldsymbol{z})$ since it has mean zero. With these terms removed, we have the path derivative estimator of the gradient that does not require the normalizing constant:

$$\hat{\nabla}_{\text{PD}}(\epsilon, \phi) = \nabla_{\boldsymbol{z}}[\log p_\theta(\boldsymbol{x}, \boldsymbol{z}(\phi, \epsilon)) - f_\phi(\boldsymbol{z}(\phi, \epsilon))]\nabla_\phi \boldsymbol{z}(\phi, \epsilon). \tag{9}$$

This estimator has zero variance when $p_\theta(\boldsymbol{z} \mid \boldsymbol{x}) = q_\phi(\boldsymbol{z} \mid \boldsymbol{x})$ and favorable performance [6]. It may be surprising that we can compute low-variance gradients without the normalization constant. However, inspection of the pathwise gradient tells we can do just that since the path through $\boldsymbol{z}$ via slice sampling does not depend on the normalization constant.

Unfortunately, evaluation of the full ELBO *does* require the normalization constant of the approximate posterior, which is no longer available.

## C  Additional experiment details

### C.1  Bias and variance experiments

Before describing specific details of each of the experiments, we first describe the reparameterized Langevin dynamics sampler. For an unnormalized distribution $p_\theta(\boldsymbol{x}) \propto \pi_\theta(\boldsymbol{x})$, the Langevin dynamics update without an accept-reject step is given by

$$\boldsymbol{x}_{n+1} = \boldsymbol{x}_n + 0.5\, \alpha^2\, \nabla_{\boldsymbol{x}} \pi_\theta(\boldsymbol{x}) + \epsilon, \quad \epsilon \sim \mathcal{N}(0, \alpha^2 I) \tag{10}$$

where $\alpha$ is a step size parameter. Notably, if $\nabla_{\boldsymbol{x}} \pi_\theta(\boldsymbol{x})$ is differentiable with respect to $\theta$, then given a realization of the random variate $\epsilon$ the update is also a deterministic and differentiable function of $\theta$. Accordingly, we can naively implement reparameterized Langevin dynamics by calling standard automatic differentiation tools on a series of Langevin dynamics updates given input randomness $\epsilon$.

Generally, Langevin dynamics sampling requires tuning the step size parameter $\alpha$ to the target distribution. We therefore expect the performance of reparameterized Langevin dynamics gradients to also be sensitive to the value of $\alpha$.

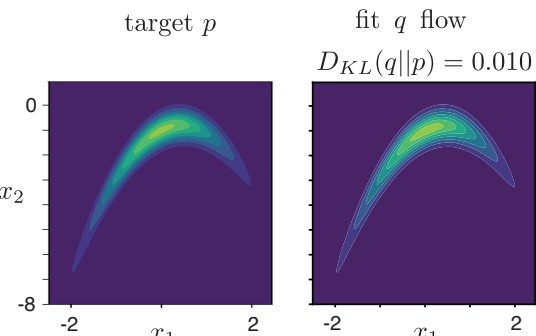

Figure 1: The target $p$ (*left*) and fit normalizing flow(*right*) approximate posterior. Each panel is shown with the same color scale.

We next describe the experiments for each of the different target distributions. In all cases, the gradient of interest is $\nabla_\theta \mathbb{E}_{q(\boldsymbol{x};\theta)}\left[\frac{1}{D}\sum_{i=1}^{D}(\boldsymbol{x}_i-k)^2\right]$ for various values of $k$, following Mohamed et al. [3]. For each Monte Carlo gradient estimator, the mean and variance of the estimated gradient was computed across 100,000 independent samples. The bias was computed by comparing the estimated mean with the known true gradient.

**1D Gaussian**  Here the target distribution is $q(x;\theta) = \mathcal{N}(x;\theta,1)$ with $\theta = 1$. The slice sampling reparameterization gradients were computed based on the loss of the $S = 5$-th, with the gradient backpropagated to the initial sample. The chain was initialized with a sample from a standard normal distribution. For this example, the estimated gradients were computed as an average across two gradient estimates. We did this so that we could compare with score function gradients calculated as a covariance across samples from slice sampling (see section E), although ultimately we did not show these gradients on the plots. We did not compare to reparameterized Langevin dynamics here, as the Langevin dynamics is motivated for choosing the appropriate direction and this is more appropriate in higher dimensions.

**5D Gaussian (1)**  In this example the distribution is 5 dimensional with $q(\boldsymbol{x};\theta) = \mathcal{N}(\boldsymbol{x};\theta\boldsymbol{1},\Psi)$ and $\Psi_1$ randomly drawn close to identity. The mean parameter was $\theta = 1$. In each of the the multidimensional experiments, we computed slice sampling reparameterization gradients using chains of length $S = 10$ or $S = 20$ steps. We estimated reparameterized Langevin gradients using $S = 10$ steps and with three step size parameters: $\alpha = 0.5, 1.0, \sqrt{2}$. In all cases, we computed the gradient based on the loss of the final sample and the Markov chains were initialized from a standard normal distribution.

**5D Gaussian (2)**  The details of this example are the same as the previous example except that the variances $\Psi_2$ were increased.

**5D Laplace**  Here the distribution was given by a 5 dimensional product of independent Laplace distributions with the same mean and different scales, $q(\boldsymbol{x};\theta) = \prod_i q(\boldsymbol{x}_i;\theta) = \text{Laplace}(\boldsymbol{x}_i;\theta,b_i)$.

### C.2   Additional details for "banana" experiment

The target "banana" distribution is

$$p(\boldsymbol{z}) = \mathcal{N}\left(\begin{bmatrix} \boldsymbol{z}_1 \\ \boldsymbol{z}_2 + \boldsymbol{z}_1^2 + 1 \end{bmatrix} \middle| \boldsymbol{0}, \begin{bmatrix} 1 & 0.9 \\ 0.9 & 1 \end{bmatrix}\right). \tag{11}$$

The neural network defining the energy had 2 hidden layers of 256 units and leaky ReLU nonlinearities. To minimize the KL divergence between the EBM approximate posterior and the target, at each iteration of training we simulated $N = 5$ independent chains each with $S = 10$ steps. We computed the loss on the final sample of each chain and backpropagated through the entire chain.

As a baseline comparison, we also fit a normalizing flow model to approximate the target distribution. We used a series of 5 masked autoregressive flow blocks [4]. The flow was implemented using the

library `jax-flows`.[1] As expected, the fit model approximates the target distribution well. It achieves slightly lower KL-divergence then the EBM approximate posterior.

## C.3  Variational contrastive divergence

We also optimized the VCD objective using with the score gradient estimator and control variates as described in Ruiz and Titsias [7], with the samples generated from slice sampling. The architecture and sampling procedure was identical to our experiments optimizing the VCD using fully reparameterized gradients. However, the test log likelihood for the fit models was substantially worse than the models optimized using reparameterization gradients (Figure 2).

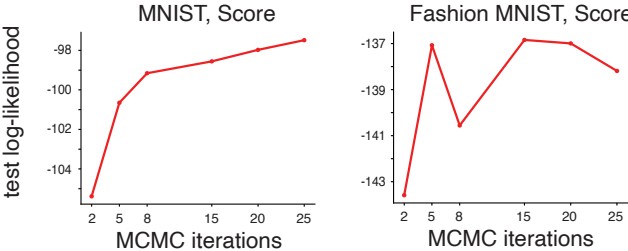

Figure 2: Test log-likelihoods when optimizing the VCD objective using the score function gradient estimator with control variates to optimize the expectation of samples targeting the true posterior. The samples were simulated via the same slice sampler that was used to compute reparameterization gradients, with different random seeds.

## C.4  VAE with Conditional EBM approximate posterior

For the VAE with a conditional EBM approximate posterior, the observations were modeled with a Bernoulli log likelihood. The latent dimensionality was $D = 20$ and the embedding dimensionality was $D_e = 50$. The gradients were estimated over mini batches of size $64$ with one MCMC chain of $S = 100$ samples for each image. We ran the optimization for 200 epochs.

We also optimized the same model using reparameterization gradients estimated from samples generated via Langevin dynamics. We set $\alpha = 1.0$ and used a noise standard deviation of $\sigma = 0.01$. Samples from the resulting model and the embedding of held-out images are shown in Figure 3. The held-out images primarily cluster into separate digit identities, and the resulting embedding is qualitatively similar to the model fit via slice sampling reparameterization gradients. However, unconditional prior samples from the model fit with reparameterized Langevin dynamics are qualitatively poor.

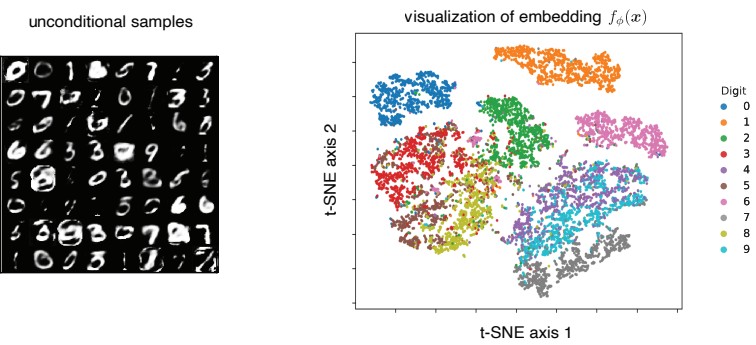

Figure 3: Unconditional samples (*left*) and embedding visualization (*right*) of a VAE fit to MNIST with an EBM approximate posterior using reparameterized Langevin dynamics.

---

[1]https://github.com/ChrisWaites/jax-flows

## C.5 Adversarial training of EBMs

Both the generator and discriminator are neural networks that map from $\mathbb{R}^2$ to $\mathbb{R}^1$ with three hidden layers of $512$ units and ReLU nonlinearities. We set the gradient penalty $\lambda = 1$.

## C.6 Bayesian sensitivity analysis

The baseball model in the main text has four hyperparameters governing the uniform distribution over $\phi$ and the Pareto distribution over $\kappa$. Three of these parameters govern the support of the distribution, so we focus on the shape hyperparameter $\alpha$ of the Pareto distribution. This

Table 1: Gradient estimates.

| method | $\nabla_\alpha \mathbb{E}_{p_\alpha}[\phi]$ | $\nabla_\alpha \mathbb{E}_{p_\alpha}[\phi^2]$ | lpd |
|---|---|---|---|
| Finite difference | 3.5174e-3 | 1.981e-3 | -1.4836 |
| Slice reparam | 3.5171e-3 | 1.981e-3 | -1.4836 |

value is set to $\alpha = 1.5$ in [2], and we evaluate the local sensitivity at this point. We first estimate the sensitivity of the first two moments of $\phi$ to the hyperparameter $\alpha$: $\frac{d}{d\alpha}\mathbb{E}_{p_\alpha}[\phi]$ and $\frac{d}{d\alpha}\mathbb{E}_{p_\alpha}[\phi^2]$. Our results are computed from 20,000 samples from the posterior. Empirical marginal distributions for two of the quantities, $\mathrm{logit}(\phi)$ and $\log \kappa$ are shown in Figure 4. We compare the slice sampling reparameterization gradient estimate with a finite difference gradient estimate. The finite difference estimate is computed by re-running the simulated samples using a fixed random seed and perturbed values for $\alpha$. The gradient estimates using the two methods match very closely (Table 1).

We also estimate the gradient of an additional quantity that is a function of the posterior samples, the pointwise log predictive density (lpd) [8]

$$\sum_{j=1}^n \log\left(\frac{1}{S}\sum_{s=1}^S p(\hat{y}_j \mid \theta^s)\right). \tag{12}$$

This is computed for heldout data $\hat{y}_j$ and $\hat{K}_j$, which are the remaining numbers of successes and trials over the course of the season for each of the players, and for $S$ samples $\theta^s$ from the posterior. The reparameterized gradient estimate for $\alpha$ matches the finite different gradient, and indicates that decreasing $\alpha$ will increase the predictions on the heldout data.

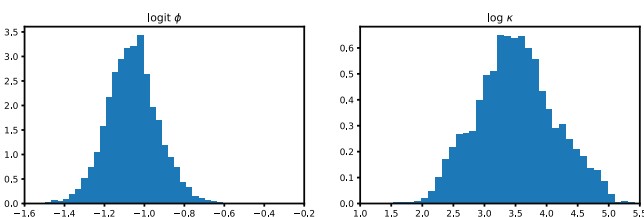

Figure 4: Marginal distributions of posterior samples for two transformed variables, $\mathrm{logit}\,\phi$ and $\log \kappa$.

## C.7 Additional minimizing KL divergence experiments

Here we show additional examples of minimizing the KL divergence between a parameterized distribution $q$ and a target $p$. In each case, the objective is $\min_\phi D_{KL}(q_\phi\|p)$ for various choices of $q$ and $p$.

**Multivariate Gaussian** First, we optimize $\min_\phi D_{KL}(q_\phi\|p)$ where $q$ and $p$ are both multivariate Gaussians with diagonal covariances. First, we show that the parameters $\phi$ optimized with slice sampling reparameterization gradients converge to the target values for a low-dimensional example with $D = 5$ (Figure 5). Next, for $D = 20$ we compare performance with different chain lengths $S$. The gradient is computed by backpropagating the loss of the final sample. As $S$ approaches $D$, slice sampling reparameterization gradients perform competitively with standard reparameterization gradients. In this experiment we also compared to the sticking the landing (STL) gradient estimator [6]; it appears that both slice sampling reparameterization gradients and the STL gradients have lower variance than the standard reparameterization gradients for Gaussian distributions.

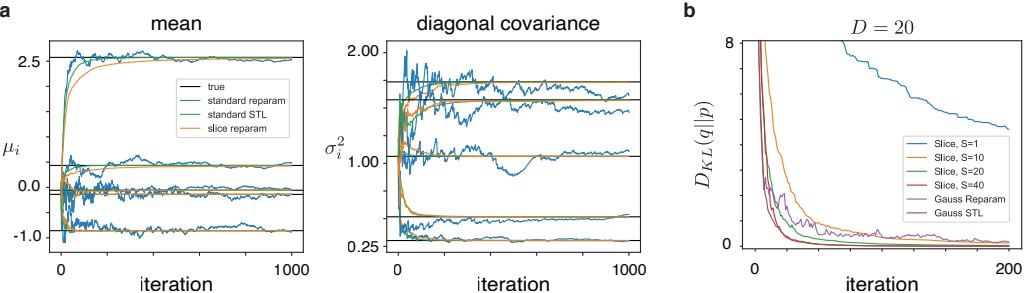

Figure 5: Matching an independent Gaussian. (**a**) Mean and diagonal covariance elements as a function of iteration. (**b**) Loss as a function of iteration for slice sampling reparameterization gradients computed using different chain lengths $S$.

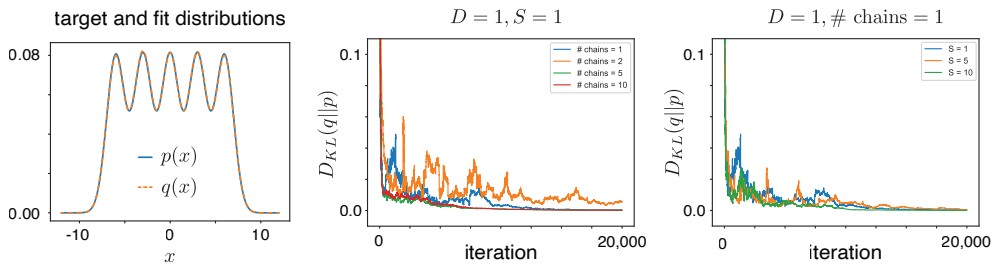

Figure 6: Fitting a mixture of Gaussians with variable numbers of samples and chains.

**Mixture of Gaussians**   Next, we minimize $D_{KL}(q_\phi||p)$ with respect to $\phi$ where $q$ and $p$ are 1D mixtures of Gaussians (Figure 6). We estimated gradients using $N$ independent chains each of length $S$ samples. While performance is reasonable with $S = 1$ and $N = 1$, increasing the number of independent chains and the number of samples improved performance. In conclusion, for multimodal distributions it may be advantageous to run multiple independent chains or increase the number of samples beyond the dimensionality of the space.

**Simulated inference network**   We implemented the simulated inference network example from Rainforth et al. [5], Section 4. In this example the generative model is

$$\boldsymbol{z} \sim \mathcal{N}(\mu, I), \quad \boldsymbol{x} \mid \boldsymbol{z} \sim \mathcal{N}(\boldsymbol{z}, I) \tag{13}$$

where $\boldsymbol{x} \in \mathbb{R}^D$ are the observations and $\boldsymbol{z} \in \mathbb{R}^D$ are the latent variables. We fit the model by optimizing the ELBO with approximate posterior $q(\boldsymbol{z} \mid \boldsymbol{z}) = \mathcal{N}(A\boldsymbol{x} + b, \frac{2}{3}I)$. The parameters are the generative parameters $\mu \in \mathbb{R}^D$ and the inference network parameters $A \in \mathbb{R}^{D \times D}$ and $b \in \mathbb{R}^D$. For a dataset $\{\boldsymbol{x}\}_{1:N}$, the optimal parameters [5] are

$$\mu^* = \frac{1}{N} \sum_{i=1}^N \boldsymbol{x}_i, \quad A^* = I/2, \quad b^* = \mu^*/2. \tag{14}$$

We simulated $N = 1000$ samples from this model and fit the generative and inference network parameters using slice sampling reparameterization gradients. The dimensionality was $D = 20$, the mini-batch size as 128, and we computed the slice sampling gradient using $S = 20$ samples after a 30 sample burn in. All parameters (true and fit) were initialized from standard multivariate Gaussian distributions. After optimizing for 1000 iterations, the fit parameters were very close to the optimal values (Figure 7).

## D   Efficient computation

Importantly, we can compute $\nabla_\theta L$ without ever fully representing either of the two Jacobians $\mathcal{J}_{\boldsymbol{x}_n}(\boldsymbol{x}_{n+1})$ and $\mathcal{J}_\theta(\boldsymbol{x}_n)$. In the reverse mode gradient accumulation, the Jacobian $\mathcal{J}_{\boldsymbol{x}_n}(\boldsymbol{x}_{n+1})$ is

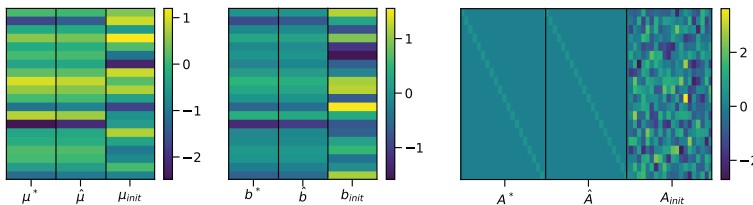

Figure 7: True ($*$), fit (), and initialized ($init$) parameters from the simulated inference network experiment [5].

always transposed and then post-multiplied by the gradient vector $\nabla_{\boldsymbol{x}_{n+1}}L$. Using the above formula for the Jacobian, this product is

$$[\mathcal{J}_{\boldsymbol{x}_n}(\boldsymbol{x}_{n+1})]^{\mathsf{T}}\nabla_{\boldsymbol{x}_{n+1}}L = [\boldsymbol{I} + u_2\boldsymbol{d}\nabla_{\boldsymbol{x}_n}[\alpha^+]^{\mathsf{T}} + (1-u_2)\boldsymbol{d}\nabla_{\boldsymbol{x}_n}[\alpha^-]^{\mathsf{T}}]^{\mathsf{T}}\nabla_{\boldsymbol{x}_{n+1}}L \tag{15}$$

$$= \nabla_{\boldsymbol{x}_{n+1}}L + u_2\nabla_{\boldsymbol{x}_n}[\alpha^+](\boldsymbol{d}^{\mathsf{T}}\nabla_{\boldsymbol{x}_{n+1}}L) + (1-u_2)\nabla_{\boldsymbol{x}_n}[\alpha^-](\boldsymbol{d}^{\mathsf{T}}\nabla_{\boldsymbol{x}_{n+1}}L). \tag{16}$$

Computing the products right to left only involves manipulating vectors. Next, the transpose of the Jacobian $\mathcal{J}_\theta(\boldsymbol{x}_n)$ is also always post-multiplied by $[\nabla_{\boldsymbol{x}_n}L]^{\mathsf{T}}$ such that

$$[\mathcal{J}_\theta(\boldsymbol{x}_n)]^{\mathsf{T}}\nabla_{\boldsymbol{x}_n}L = [u_2\boldsymbol{d}\nabla_\theta[\alpha^+]^{\mathsf{T}} + (1-u_2)\boldsymbol{d}\nabla_\theta[\alpha^-]^{\mathsf{T}})]^{\mathsf{T}}\nabla_{\boldsymbol{x}_n}L \tag{17}$$

$$= u_2\nabla_\theta[\alpha^+](\boldsymbol{d}^{\mathsf{T}}\nabla_{\boldsymbol{x}_n}L) + (1-u_2)\nabla_\theta[\alpha^-](\boldsymbol{d}^{\mathsf{T}}\nabla_{\boldsymbol{x}_n}L). \tag{18}$$

# E   Score function gradients as covariance

Let $q(\boldsymbol{x};\theta)$ be a distribution with an unknown normalization constant

$$q(\boldsymbol{x};\theta) = \frac{1}{Z(\theta)}e^{f_\theta(\boldsymbol{x})}. \tag{19}$$

The score function gradient requires $\nabla_\theta \log q(\boldsymbol{x};\theta)$, which cannot be directly computed. This leads to an alternative formulation of the score function gradient, written as a covariance between the cost function $g(\boldsymbol{x})$ and the gradient of the unnormalized log energy $f_\theta(\boldsymbol{x})$:

$$\begin{aligned}
\nabla_\theta\mathbb{E}_{q(\boldsymbol{x};\theta)}[g(\boldsymbol{x})] &= \mathbb{E}_{q(\boldsymbol{x};\theta)}[g(\boldsymbol{x})\nabla_\theta \log q(\boldsymbol{x};\theta)] \\
&= \mathbb{E}_{q(\boldsymbol{x};\theta)}[g(\boldsymbol{x})(\nabla_\theta f_\theta(\boldsymbol{x}) - \nabla_\theta \log Z(\theta))] \\
&= \mathbb{E}_{q(\boldsymbol{x};\theta)}[g(\boldsymbol{x})\nabla_\theta f_\theta(\boldsymbol{x})] - \mathbb{E}_{q(\boldsymbol{x};\theta)}[g(\boldsymbol{x})]\nabla_\theta \log Z(\theta) \\
&= \mathbb{E}_{q(\boldsymbol{x};\theta)}[g(\boldsymbol{x})\nabla_\theta f_\theta(\boldsymbol{x})] - \mathbb{E}_{q(\boldsymbol{x};\theta)}[g(\boldsymbol{x})]\mathbb{E}_{q(\boldsymbol{x};\theta)}[\nabla_\theta f_\theta(\boldsymbol{x})] \\
&= \mathrm{Cov}_{q(\boldsymbol{x};\theta)}[g(\boldsymbol{x}), \nabla_\theta f_\theta(\boldsymbol{x})].
\end{aligned}$$