# OpenReview forum: "Slice Sampling Reparameterization Gradients"
_NeurIPS.cc/2021/Conference — NeurIPS 2021 Spotlight_

### Official Review · Reviewer_dDr8 · 2021-07-15

**Rating:** 7
**Confidence:** 3

**Summary:**

The authors present a slice sampling technique for computing reparameterized gradients for unnormalized distributions. They do this by using a deterministic version of slice sampling by computing the end-points of the slice via a root finding procedure that makes the sampling bath deterministic and differentiable. The authors provide comprehensive experimental results in support of the technique on a variety of problems.

**Limitations And Societal Impact:**

The authors haven't included a discussion around the limitations of the proposed method. At the least I would consider including a discussion around the *possible* limitations for finding the exact end-points and the computational cost especially when longer chains are need to simulate samples from the target density.

**Main Review:**

I found the general idea proposed in the paper by the authors well motivated and interesting. The main idea has been presented with much clarity by the authors and the intuitive explanations around the procedure followed by mathematical details is most welcome (and refreshing; I wish more papers follow such practice). The idea itself is sufficiently novel and empirically back-up by the experiments by the authors. The experiments are nicely designed to give nice proofs-of-concept for the method.

A small nit I'll add wrt to the experiment in section 4.3 and Fig 4: If possible, it will be nice if the left figure in Fig 4 has the baseline for VCD with architecture similar to that in [45] as well (instead of being refered to another paper). This will make the comparison immediately obvious and I believe the authors can generate that plot easily considering they use the same setup as in [45].

I am also curious about the cost of finding the end-points in general. On surface, it seems this might become trickier (especially in higher dimensions) and wanted to know if the authors encountered any problems around it in their experiments. I am also curious how this whole pipeline of finding the exact endpoints etc effects the running time in general.


**Time Spent Reviewing:**

~6

---

> ### Author Response · Authors · 2021-08-10
> **Response to Review of Paper6680 by Reviewer dDr8**
>
> We appreciate the reviewer’s careful evaluation of the paper.
>
> *1. Cost of finding endpoints.* Notably, the cost of finding the slice endpoints is approximately the same regardless of the dimensionality of the sample space, as even in higher dimensions we only ever search for the slice endpoints along a single direction. Rather, the cost of increasing dimensions arises through the number of samples required to get a “good” sample. Overall, identifying the endpoints requires function evaluations both for the root finding algorithm (here we used a standard bisection algorithm) to identify the root and for our procedure to choose the initial bracketing endpoint. Finding the endpoints becomes more challenging for multi-model distributions with closely grouped or quickly varying modes, since this would require a finer search space to identify the nearest point that satisfies the slice boundary condition. Therefore, the primary factors that increase overall computational complexity are the total number of samples, the cost of function evaluation, and the number of function evaluations to identify the endpoints.
>
> An example of the computational cost is as follows. For section 4.2, we trained the EBM approximate posterior with 30K iterations of SGD. At each iteration we computed the loss on the final samples from 5 chains of 10 samples each. For our setup running on a cluster with a single GPU node, each iteration took about 50ms (~20 iters / second) such that the overall training was about 24.5 minutes. We thank the reviewer for asking about the cost of finding the slice endpoints, and we will include these details in a more clear and thorough discussion of the cost in the revised manuscript.
>
> *2. VCD baselines.* Thank you for your suggestion to add the VCD baselines from [45] to the left figure in Fig 4. We agree that it would be worthwhile to implement the VCD baseline to compare the exact numbers across different sample sizes. We will implement the alternative gradient estimator from [45] and rerun the experiments using that estimator to make a direct comparison.
>
> *3. Limitations.* We agree that discussing the limitations of the method --- when we expect it to be difficult or expensive to find the slice endpoints or the method to require prohibitively long chains --- is important. We thank the reviewer for bringing this up and will include a more clear discussion in the revised manuscript.

---

### Official Review · Reviewer_915a · 2021-07-16

**Rating:** 6
**Confidence:** 4

**Summary:**

This paper introduces an implementation of slice sampling that is compatible with automatic differentiation. This allows differentiation through slice sampling trajectories in probabilistic loss functions when learning generative models. The authors implement their method for different generative learning tasks to show its utility.

**Limitations And Societal Impact:**

The authors have adequately address the limitations and societal impact of their work.

**Main Review:**

The main contributions of this work are 1) introduction of an efficient slice sampling implementation that is compatible with automatic differentiation and 2) a reparameterization method for evaluating gradients through probabilistic loss functions directly (i.e. without the need for MCMC samples).

The slice sampling method that the authors base their work on is classical and the central innovation for making it differentiable is ensuring the differentiability of the process of finding the local "width" of a given slice. This is a nice idea. One major drawback of their sampling method is the lack of information about the landscape geometry when choosing the direction for MCMC updates, since this is chosen uniformly. The standard method Langevin sampling uses the gradient term to guide is trajectories, which can be helpful for exploring high dimensional spaces. I am not sure MCMC methods based on random search directions can scale as well, and it might be interesting for the authors to consider improving this part of their algorithm.

The reparameterization gradient method introduced in this paper is the more interesting contribution from my perspective. Instead of simply using differentiable slice sampling as a tool for evolving MCMC samples, the authors find an efficient method for differentiating through the slice sampling update process to get loss gradients directly. One part of the method raises some questions. It appears that in the explanation of (7), the authors are using a sampling trajectory $(x_0, x_1, ... , x_N)$ as samples for approximating an expectation with the law of large numbers. This does not seem appropriate, because the samples will be highly correlated since they belong to the same trajectory. It would make more sense to me to use the last sample $x_N$ from independent chains. Unless I am mistaken, it appears that slice reparameterization uses a single chain to approximate the expectation. Further clarification of this would be helpful.

The experimental results demonstrate a reasonably wide variety of applications for the proposed method. While the scale and complexity of the experiments is somewhat limited, the proposed method seems to work well in different situations. Nonetheless, there doesn't seem to be any significant benefit to the slice sampling as opposed to Langevin sampling except for more straightforward tuning.

I feel like too much space is devoted to experimental results and not enough space is devoted to demonstrating how the reparameterization method works in practice. The authors give a brief derivation of the ELBO reparamterization in the paper and put the rest in the appendix. As far as I can see, there is no derivation of reparameterization of the KL loss for EBM learning, and it is not immediately clear to me how this is done (I would assume the reparameterization trick is used on the term $E_{q_\phi} [ \nabla_\phi f_\phi (X) ]$ term from the derivative of the log normalizer). Since the experimental results travel well-established ground in terms of the kinds of experiments presented, the paper would be strengthened by focusing more on the details of the new method.

**Overall**: This paper presents some nice ideas for introducing a new slice sampling method and for absorbing sampling trajectories directly into the loss function in generative modeling. More complex experiments (color images perhaps) could help strengthen the paper. There are a few points where further discussion would help strengthen my understanding of the the proposed method.

**Time Spent Reviewing:**

5 hours

---

> ### Author Response · Authors · 2021-08-10
> **Response to Review of Paper6680 by Reviewer 915a**
>
> We appreciate the reviewer’s careful evaluation of the paper.
>
> *1. Scalability of random directions.* The reviewer is correct that the random direction slice sampling algorithm used in this paper does not incorporate gradient information, which is a limitation for higher dimensional spaces. Effectively including gradient information in slice sampling is an important open research question and we are interested in pursuing this in the future. We will include a discussion of this limitation in the main text. Importantly, we still expect to use the proposed approach in problems with tens of dimensions, where there are interesting problems and for which our experiments have shown that the algorithm can be useful.
>
> *2. Langevin dynamics.* Since Langevin dynamics incorporates gradient information, we also expect it to explore the sampling space better in high-dimensional spaces such as images. However, as the reviewer states, one drawback of Langevin dynamics is the requirement of (often) careful tuning of the step size parameters. In practice, we believe this is not a trivial drawback, as this tuning can be challenging and it can be difficult to know when a good tuning has been achieved. Additionally, it is worth noting that Langevin dynamics without an accept-reject step targets an asymptotically biased posterior distribution; adding a Metropolis-Hastings correction leads to non-differentiability and therefore cannot be used for reparameterization. On the other hand, slice sampling in theory targets the true posterior. Therefore, gradient estimates from reparameterized slice sampling are in theory asymptotically  unbiased while those from Langevin sampling are asymptotically biased. Both slice sampling and Langevin dynamics have initialization bias, as all MCMC methods do. We will improve our comparison of these methods in the manuscript.
>
> *3. Samples used in evaluating loss.* We thank the reviewer for asking about what samples are used for computing the loss. We presented the formulation in equation (7) with full generality, where there can be a loss computed on every sample. However, it is certainly the case that we can only compute the loss on a subset of samples, such as just on the last sample. We used a variety of settings in our experiments. Typically, we used multiple independent chains and computed the loss based on the final sample or multiple samples. In the VAE experiments, we used multiple independent chains where each chain targets an observation-specific distribution.
>
> There are arguments for both using multiple independent chains and for using one longer chain. Using multiple samples from one longer chain increases variance but generally decreases bias due to the initialization. One primary benefit of using multiple independent chains in parallel is computational speed. And while each chain would have more initialization bias than one longer chain, it is possible that the multiple independent chains from different initializations would have explored different areas of the distribution better (e.g., different samples explore different modes). For these reasons, when possible we often used multiple independent chains in parallel. In the revised paper, we will make clear what samples we used to evaluate the loss and how many chains we used.
>
> *4. Space of experimental results.* We appreciate the reviewer noting that some details about the method are unclear, and that devoting more space to the method rather than experiments would improve understanding in clarity. We will make this change in the revised paper and will make sure to specify how we reparameterize each objective (including the KL loss).

---

> > ### Comment · Reviewer_915a · 2021-08-19
> > **Thanks for the clarifications**
> >
> > I appreciate the author's thoughtful responses. In my view, the most significant problem for MCMC samplers in high dimensions is finding meaningful proposal directions (directions along the local geometry of the manifold of observed) and I am not sure that this sampler fully addresses this central issue. Nonetheless, there are a relatively limited selection of choices for MCMC samplers for deep learning and I appreciate the author's efforts to introduce this new technique. After reading the responses of other reviewers and considering the authors response, I am changing my score. For this method to reach its fully potential, I believe that future work should focus on incorporating useful information from the energy surface.

---

### Official Review · Reviewer_SVGs · 2021-07-16

**Rating:** 7
**Confidence:** 3

**Summary:**

This paper provides a method based on automatically differentiable slice-sampling to compute gradients of an expectation with respect to parameters of the distribution, even when the normalization constants of the respective distribution are unknown. In such cases, the proposed method provides a low-variance (as opposed to score-gradients) alternative to reparameterization gradients.

**Limitations And Societal Impact:**

The limitations of the work are unfortunately not discussed. For example, the runtime in comparison to other methods is not investigated. Discussion of limitations should be extended. There is no immediate societal impact.

**Main Review:**

Overall, the paper is well written and the presentation is clear. The proposed method is clearly useful and the experiments support this further since you 1) compare to other gradient estimators and 2) present experiments with EBMs with unavailable normalization constants. The solution with slice sampling and the observation that the slice sample path is differentiable is very elegant.

### Comments and questions
- are there cases where finding the slice boundaries is very inefficient or inexact? What happens in this case? Certainly the samples are biased (i.e. not from the target distribution). Is this ever a problem in practice?
- Figure 2 is quite hard to parse. For example, the caption could mention that the figure titles correspond to the distribution $q(x;\theta)$ and the parameter $\theta$ at which the gradient is estimated. This is made clear in 4.1 which is significantly more verbose. Another suggestion is to highlight the proposed method *slice $S=10$* and *slice $S=20$*, e.g. by using a different linestyle or thickness for the corresponding lines. This allows to quickly gauge the performance of the approach and understand the impact of $S$.
- Section 4.2 (toy example) suggests at first that the proposed method combined with EBMs is the only method that could achieve such a fit but I suppose methods like normalizing flows could achieve a similar distribution, right? It would be good to mention something like this, or that the goal is just to show what's possible when using the method with EBMs that would be hard otherwise.
- Section 4.4: which other methods would enable this application? If there is no other method that could compete here, state and explain it, otherwise I would suggest to have some sort of baseline here. Surely, the application is interesting and impressive but hard to relate to existing methods.
- I would be interested see the comparison of runtime for the experiment in Figure 2 to understand how expensive slice samples are. From what I expect, the root finding could take long, especially in higher dimensions. Do you have some results on this?
- Overall, the experiments section shows the proposed method in a variety of settings and also on a breadth of problems which is nice. However, I think it is important to explain why the proposed method should be used instead of other possible approaches and how these alternative approaches would perform in comparison to your method. If no other method is applicable in these cases, this can be stated and explained. Maybe you could elaborate a bit on the lack of baselines/methods you compare to?

### Typos
- line 215 *..variance the score..* -> *..variance than score..*?



**Time Spent Reviewing:**

5

---

> ### Author Response · Authors · 2021-08-10
> **Response to Review of Paper6680 by Reviewer SVGs**
>
> We appreciate the reviewer’s careful evaluation of the paper.
>
> *1. Finding slice endpoints.* There are cases where finding the slice endpoints can be inefficient or inexact. We need to identify the closest point that satisfies the slice boundary condition. Our procedure for this is described in Suppmat Sec. E, Lines 135-141. We first “step out” from the current point until a bracketing point is found before running the root finder algorithm. The hope is that this bracket contains only one point that satisfies the boundary condition, which is the slice endpoint. However, for multimodal distributions we cannot in general guarantee we have only bracketed the true slice endpoint, and therefore finding the slice boundaries can be inexact for multimodal distributions. In particular, this is more challenging when modes are closely connected such that there are multiple nearby points that satisfy the slice endpoint condition. To verify if this is an issue, we can check if the next sampled point is on the slice. And if it is an issue, we can modify the stepping out procedure to probe more finely spaced locations.
>
> The reviewer is correct that if the endpoints are not correctly identified then the resulting gradient estimates will have asymptotic bias. We therefore also expect learning to be poor in that case. Our observation of this motivated the “step out” procedure to choose the initial bracket, which was important for getting good performance on multimodal examples (e.g., Sec. 4.5 Adversarial training of EBMs). We will include this discussion in the revised manuscript.
>
> *2. Figure 2 clarity.* We will improve the clarity of Figure 4.2 using consistent colors for each method in the revised paper.
>
> *3. Section 4.2.* We agree that other methods that can approximate complex distributions such as normalizing flows would likely also perform well in this experiment. As the reviewer suspected, our intention was to motivate the proposed reparameterization gradient method that applies to a general set of distributions including EBMs, and not to imply that other methods could not achieve comparable performance in this experiment. We will make the motivation for this experiment and all experiments more clear in the revised paper.
>
> *4. Section 4.4.* Other ways to train a general EBM approximate posterior would be to use score function gradients or reparameterized Langevin dynamics. Next, another way to train a more flexible posterior would be to use normalizing flows. We will include 1) an EBM approximate posterior trained with reparameterized Langevin dynamics or score function gradients, and 2) a normalizing flow baseline in the revised manuscript.
>
> *5. Runtime.* For Figure 4.2, since the EBM has a neural network energy function it takes significantly longer to train than the 2D Gaussian distribution in this case. We trained the EBM approximate posterior with 30K iterations of SGD. At each iteration we computed the loss on the final samples from 5 chains of 10 samples each. For our setup running on a cluster with a single GPU node, each iteration took about 50ms (~20 iters / second) such that the overall training was about 24.5 minutes. On the other hand, the 2D Gaussian distribution has only 4 parameters and converges with fewer iterations.
>
> Notably, the cost of generating a sample is similar in higher dimensions. We only search for the slice endpoints along a single direction, so the procedure remains a one dimensional problem even in multidimensional spaces. Rather, the computational costs scale linearly with the total number of samples generated, which will generally increase for higher dimensional spaces.
> We will include these details in the revised paper.
>
> *6. Baselines.* For each experiment, we will add a discussion of possible alternative approaches and how they would compare to reparameterized slice sampling gradients. We will also add one or more baseline experiments (See Point 4).
>
> *7. Limitations.* The primary limitations are that the method may require prohibitively long chains in some cases and that it may be difficult or expensive to find the slice endpoints (as described above). We will make sure to include a discussion of the limitations in the revised paper.

---

### Official Review · Reviewer_Pwj1 · 2021-07-19

**Rating:** 7
**Confidence:** 4

**Summary:**

The authors propose a technique to compute reparameterized gradients w.r.t. parameters of an unnormalized variational density based on slice sampling.
Instead of performing slice sampling with the common "stepping out & shrinkage" procedure the authors use a numeric root finding procedure to determine the slice endpoints, which does not require accept/reject steps and is therefore differentiable.


**Limitations And Societal Impact:**

adequate

**Main Review:**

**Summary:**
The authors propose a technique to compute reparameterized gradients w.r.t. parameters of an unnormalized variational density based on slice sampling.
Instead of performing slice sampling with the common "stepping out & shrinkage" procedure the authors use a numeric root finding procedure to determine the slice endpoints, which does not require accept/reject steps and is therefore differentiable.

**Originality:**
The presented work is novel and presents an interesting new combination of (auxiliary-variable) MCMC and (stochastic) optimization/variational inference.

**Clarity:**
The paper is clearly written and easy to follow given it's technical depth. The authors do a good job in explaining the relevant techniques along the way without getting side-tracked.

**Quality:**
This is a high quality paper which is reflected in the presentation, writing.
However, with the exception of experiment 4.3, the authors do not evaluate against other suitable state-of-the-art methods nor baseline. Hence, while the authors demonstrate that their method can be applied and provides useful results for a variety of tasks, it is impossible to assess how good these results are compared to existing methods in practice.
For example, the value of the KL-divergence in experiment 4.2. seems small and the density seems to be very similar to the target density. But it is unclear if a simple flow-based model or SMC(S)-based model would not have achieved similar or better results. Based on the limited evaluation (only on rather simple target density) it's unclear what the benefits of training neural density approximation with the proposed method are.
In my opinion, the paper would benefit from focusing on fewer but more substantial experiments, i.e. experiments with a clear aim and conclusion/takeaway which demonstrate the usefulness of the proposed method compared to existing methods.
My takeaway after reading the experiments section was that the method is usable in a variety of settings, but I was not able to clearly see the benefits from using the proposed method over other, possibly simpler, existing methods.

**Significance:**
Training variational densities/energies is an important task in reinforcement learning, probabilistic machine learning and beyond. I think the paper is methodologically interesting and makes a significant technical contribution to the field. As such it is of interest to the broader NeurIPS community.
However, as described above, without a more thorough evaluation of the performance of their methods compared to other state-of-the-art methods it's hard to assess the impact/usefulness of the presented contribution in practice.

**Additional Questions and Comments:**
- I wonder if there is a case where discarding initial samples from the slice sampler, i.e. a burn in period, is useful? Is it the case that even early samples provide a useful contribution to the gradient signal and hence should not be discarded, or are there settings, e.g. highly complicated densities, where the initial samples introduce an unacceptable bias to the gradient signal.
- Typo on line 215: variance *than* the score function gradients



**Time Spent Reviewing:**

6

---

> ### Author Response · Authors · 2021-08-10
> **Response to Review of Paper6680 by Reviewer Pwj1**
>
> We appreciate the reviewer’s careful evaluation of the paper.
>
> *1. Baselines and comparison to other methods*. The reviewer notes that more baselines would be helpful and important for determining the relative performance of the models trained using the proposed method. Regarding experiment 4.2, we agree with the reviewer that e.g. a flow-based model would likely perform well in this case. Our intention with this experiment was to conceptually motivate the proposed reparameterization gradient method that applies to a general set of distributions including EBMs, and not to imply that no other methods or models would perform comparatively in this experiment. Thank you for raising this point. We will make our intentions for this experiment and the rest of the experiments more clear.
>
> We would also like to emphasize that we are proposing a new method to compute reparameterization gradients, rather than a specific model. This method may be combined with other models such as flow-based models, which we hope to do in future work. Additionally, this method also may be applicable even when other models are not. For example, in both the variational contrastive divergence (Section 4.3) and Bayesian sensitivity experiments (Section 4.6), we use slice sampling reparameterization gradients to differentiate samples from the true model posterior. To obtain reparameterization gradients in both cases, we require a method that applies to multivariate, unnormalized distributions such as the method we propose. We hope this makes the utility of the proposed method more clear.
>
> *2. Number of experiments*. It has been suggested by more than one reviewer that there are potentially too many experiments in the main text. We appreciate this comment, and we will move some experimental results and text to the supplement to create more space for elaboration on the method and primary results in the main text.
>
> *3. Discarding initial samples*. The reviewer is correct that there are certainly cases where discarding the initial samples can be useful. This is true for at least two reasons. First, running a longer chain can help improve convergence and reduce initialization bias. Second, ignoring burn in samples reduces computational costs because we do not need to backpropagate through as many samples. So we can simulate a set of “burn in” samples before simulating the samples that we differentiate to reduce estimator bias and computational requirements, when necessary. We found that it was important to do this in experiment 4.4.
>
> *4. Typo*. We will fix the typo on line 215. Thank you.

---

> > ### Comment · Reviewer_Pwj1 · 2021-08-25
> > **Response**
> >
> > Thank you for your detailed answers.
> >
> > After reading the other reviews and responses, it seems to me that the rebuttal addresses most concerns. For my part, I'm happy with the rebuttal and think that adding additional discussion and baselines to the experiment section, as promised in the response to *Reviewer SVGs*, are likely to improve the paper.
> >
> > I will increase my score, recommending to 'accept' the paper, accordingly.

---

### Decision · Program_Chairs · 2021-09-27

**Decision:**

Accept (Spotlight)

**Comment:**

This paper introduces a differentiable slice sampler and considered its application to variational inference. The reviewers unanimously agreed that this was an important contribution. One reviewer was concerned that the use of uniform search directions were a draw back of the method. Many reviewers felt that the paper could benefit from fewer, more focused experiments.

After discussion, the reviewers unanimously agreed that the paper should be accepted, as long as the authors make the changes that they propose (including additional discussions on the method and reducing the density of the experiments). I am happy to recommend acceptance.